# A Flexible Large Language Models Guardrail Development Methodology Applied to Off-Topic Prompt Detection

**Gabriel Chua**[*]
Government Technology Agency
Singapore
gabriel_chua@tech.gov.sg

**Chan Shing Yee**[†]
National University of Singapore
Singapore

**Shaun Khoo**
Government Technology Agency
Singapore

## Abstract

Large Language Models (LLMs) are prone to off-topic misuse, where users may prompt these models to perform tasks beyond their intended scope. Current guardrails, which often rely on curated examples or custom classifiers, suffer from high false-positive rates, limited adaptability, and the impracticality of requiring real-world data that is not available in pre-production. In this paper, we introduce a flexible, data-free guardrail development methodology that addresses these challenges. By thoroughly defining the problem space qualitatively and passing this to an LLM to generate diverse prompts, we construct a synthetic dataset to benchmark and train off-topic guardrails that outperform heuristic approaches. Additionally, by framing the task as classifying whether the user prompt is relevant with respect to the system prompt, our guardrails effectively generalize to other misuse categories, including jailbreak and harmful prompts. Lastly, we further contribute to the field by open-sourcing both the synthetic dataset[1] and the off-topic guardrail models[2], providing valuable resources for developing guardrails in pre-production environments and supporting future research and development in LLM safety.

## 1 Introduction

Large Language Models (LLMs) such as GPT-4o (et al., 2024b), Gemini 1.5 (et al., 2024a), and Llama 3 (Llama Team, 2024) have revolutionized various sectors by enabling advanced natural language processing capabilities. Their applications extend beyond conversational agents to include tasks such as document extraction, report generation, and workflow automation (Brachman et al., 2024). As these models become increasingly integrated into software applications and real-world processes, ensuring appropriate use is critically important.

To mitigate potential risks, significant efforts have been made to develop model alignment (Christiano et al., 2017) and guardrails (Dong et al., 2024). Alignment techniques aim to ensure that LLMs behave in accordance with human values and intentions, while guardrails are external mechanisms that prevent models from generating unwanted or harmful outputs. These safety measures are crucial to maintain user trust and meet regulatory requirements, especially in sensitive domains such as healthcare, finance, and legal services.

---

[*]Corresponding Author

[†]Work was done during an internship at the Government Technology Agency.

[1]https://huggingface.co/datasets/gabrielchua/off-topic

[2]https://huggingface.co/collections/off-topic-guardrail-673838a62e4c661f248e81a4

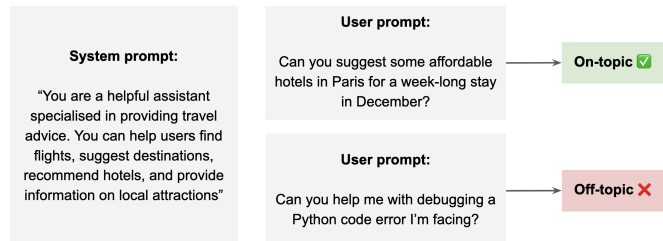

Figure 1: Example of on- and off-topic user prompts: The goal is to classify whether a user prompt is off-topic with respect to the developer-defined system prompt.

One specific challenge is off-topic misuse: Users may prompt LLMs to perform tasks outside their intended scope, sometimes unknowingly, and other times to circumvent organizational policies. For instance, a healthcare policy chatbot could generate Python code upon request. We refer to these prompts as "off-topic" (see Figure 1), which differ from "jailbreak" prompts (Shen et al., 2024) that explicitly seek harmful or disallowed content. Although off-topic prompts may be benign, they can still undermine the intended functionality and carry compliance risks (e.g., inadvertently providing medical or legal advice).

Existing guardrails often rely on curated datasets or blacklists (AWS; Azure; Rebedea et al., 2023), but these suffer from high false positives, limited adaptability, and the impracticality of gathering large real-world data in pre-production. This highlights three key challenges:

1. The need for a general-purpose, robust classifier to detect off-topic prompts,
2. The requirement to build such a classifier *without* large-scale real-world data,
3. The difficulty that real user data is typically absent before deployment.

**Contributions.** In this paper, we:

1. **Propose a Flexible, Data-Free Guardrail Development Methodology:** We detail how to generate synthetic data *in pre-production* to build and benchmark guardrails, thus providing a strong safety baseline even before real-world data is available.
2. **Develop Performant Off-Topic Guardrails:** We train simple yet effective embedding and cross-encoder classifiers that dramatically reduce false positives and achieve strong recall on multiple external benchmarks.
3. **Showcase Generalization to Multiple Misuse Categories:** Our approach effectively handles malicious prompts (e.g., jailbreaking, harmful requests) by classifying them as off-topic relative to a specialized system prompt.
4. **Open-Source the Dataset and Models:** We release both the synthetic dataset[3] and the guardrail models[4], encouraging broader adoption and further research in LLM safety.

**Paper Organization.** In Section 2, we survey the key challenges of LLM safety, synthetic data use, and relevant research gaps. Section 3 details our guardrail development framework and data generation process. Section 4 presents experimental results, including performance comparisons, ablation studies, and analyses of threshold settings and model calibration. Section 5 and 6 discusses limitations, future work and practical deployment considerations repsectively. Finally, Section 7 concludes the paper.

## 2 RELATED WORK

Ensuring safe and aligned behavior of LLMs is a critical open challenge. Below, we summarize how existing alignment and guardrail techniques are tackling these issues and highlight key unresolved

---

[3]https://huggingface.co/datasets/gabrielchua/off-topic
[4]https://huggingface.co/collections/off-topic-guardrail-673838a62e4c661f248e81a4

points—particularly, how synthetic data generation can help fill the gap when real-world data is absent.

## 2.1 ALIGNMENT CHALLENGES IN LLMS

Alignment seeks to ensure that model outputs adhere to human values and developer intentions (Christiano et al., 2017; Ouyang et al., 2024). However, even state-of-the-art approaches like Reinforcement Learning from Human Feedback (RLHF) (Rafailov et al., 2024; Lu et al., 2024) face limitations, such as over-refusal of valid queries (Ganguli et al., 2022) or misalignment on unseen tasks. Off-topic prompts can inadvertently bypass alignment constraints if the model has not been explicitly trained to detect domain or scope violations. Thus, a dedicated method for filtering out off-topic queries remains necessary, complementing alignment methods.

## 2.2 GUARDRAILS AND THEIR GAPS

Guardrails are often implemented as external classifiers or filters on top of an LLM (Rebedea et al., 2023; Inan et al., 2023). Unlike alignment, guardrails can be updated without retraining the base model, offering faster iteration and adaptation to new threats. Current practices rely heavily on curated examples (AWS; Azure) or rule-based systems. However, enumerating all off-topic or unsafe prompts manually is not feasible, and collecting real data in pre-production is often impossible.

## 2.3 SYNTHETIC DATA FOR LLM SAFETY

Synthetic data is increasingly used when real data is limited or sensitive (Liu et al., 2024). Early works leverage LLMs for pseudo-labeling (Long et al., 2024), QA or retrieval augmentation (Xu et al., 2024), and instructional dialogue generation (Wang et al., 2023). More recent efforts (Sharma et al., 2025) have demonstrated the potential of synthetic data to train content classifiers and deploying them at scale.

Despite these advances, using synthetic data *specifically* to train off-topic detectors has been underexplored. Potential challenges include ensuring coverage of diverse topics and controlling the style or format of prompts. Our work addresses this gap by proposing a systematic, LLM-based method to generate large, varied synthetic datasets for off-topic detection. This approach also generalizes to other misuse cases (e.g., harmful requests), creating a strong pre-deployment guardrail and significantly lowering the risk of over- or under-refusal.

**Unresolved Key Issue:** How can we create a sufficiently broad and representative dataset to train lightweight guardrails *before* real-world deployment? This is where synthetic data generation offers a promising solution.

## 3 METHODOLOGY

We now describe our general-purpose, data-free guardrail development framework and apply it to the specific problem of detecting off-topic prompts in LLM interactions. The framework consists of three main steps: (1) *Qualitative problem analysis,* (2) *Synthetic data generation*, and (3) *Model training*.

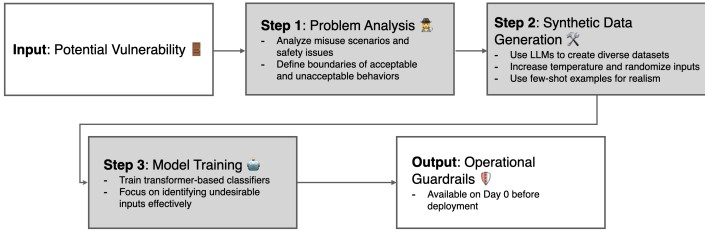

Figure 2: Our Guardrail Development Methodology

## 3.1 GUARDRAIL DEVELOPMENT FRAMEWORK

**Step 1: Qualitative Problem Analysis & Edge Case Identification.** We begin by comprehensively describing the intended model use cases and the definition of "misuse." Specifically, we consider *off-topic* to mean "Any user prompt that is irrelevant to the domain or scope specified in the system prompt." We also brainstorm potential edge cases, such as extremely short or vague prompts, potentially multi-lingual prompts, and adversarial attempts to circumvent the scope. This step ensures we have a clear, qualitative understanding of what off-topic behavior looks like.

**Step 2: Synthetic Data Generation via LLM Prompting.** Next, we employ a large language model (e.g., GPT-4o, Llama 3) to generate synthetic examples. We craft a carefully written "meta-prompt" instructing the LLM to produce a variety of (system prompt, user prompt) pairs. For each system prompt, we ask the LLM to generate both on-topic and off-topic user prompts, ensuring a balanced dataset. To promote diversity, we vary:

- The domain and style of the system prompt (healthcare Q&A, legal summary, short domain instructions, etc.),
- The complexity of user prompts (short queries, multi-sentence requests, or multilingual prompts),
- Random seed words and generation parameters like temperature and top-k sampling.

This process can easily produce thousands to millions of synthetic examples without any real user data. After generation, light heuristics or human verification can be applied to remove low-quality or duplicate examples.

**Step 3: Model Training.** We train a dedicated classifier on the synthetic data. Concretely, each training instance has:

$$(\text{System Prompt } S, \text{User Prompt } U) \rightarrow y \in \{0, 1\}$$

where $y = 1$ if $U$ is off-topic relative to $S$, and $0$ otherwise. We explore both a *bi-encoder* approach (embedding system and user prompts separately) and a *cross-encoder* approach (concatenating the prompts into a single sequence). The fine-tuning objective is standard binary classification. Crucially, the output includes a probability score, allowing us to set thresholds for refusal or escalation based on application risk tolerance.

## 3.2 OFF-TOPIC PROMPT DETECTION FORMULATION

Formally, we define a function:

$$F(S, U) \rightarrow \{0, 1\},$$

where $S$ is the system (developer) prompt defining the intended scope, and $U$ is the user prompt. If $F(S, U) = 1$, then $U$ is deemed off-topic, triggering a refusal or guidance response. In addition to the binary label, $F$ can produce a score $p \in [0, 1]$ indicating the likelihood of being off-topic, which provides a continuous trade-off between false positives and false negatives.

## 3.3 MODELING

We explore two modeling approaches (see Figure 3):

**1. Fine-Tuned Bi-Encoder Classifier** We start with a pre-trained embedding model that is lightweight and supports long sequences. Specifically, we use jina-embeddings-v2-small-en (Günther et al., 2024), which has 33M parameters and an 8k token limit. For our experiments, we trim sequences to 1k tokens.

We feed the system prompt and user prompt into the embedding model separately. We then learn "adapter" layers (and cross-attentions) on top of each embedding. Next, we use attention pooling to get a single vector representation from each branch and concatenate them. Finally, we use a classification head to determine on-topic vs. off-topic.

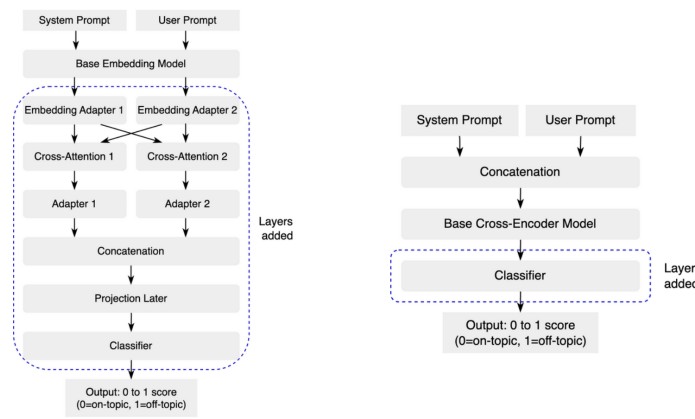



**(a) Fine-Tuned Bi-Encoder Classifier**      **(b) Fine-Tuned Cross-Encoder Classifier**



Figure 3: Summary of the two modeling approaches for off-topic prompt detection

**2. Fine-Tuned Cross-Encoder Classifier**  We also fine-tune a pre-trained cross-encoder model (e.g., stsb-roberta-base). In this approach, we concatenate the system and user prompts into a single sequence. The cross-encoder then processes this combined input. Finally, we apply a classification head to the pooled representation to yield the off-topic prediction.

# 4 EXPERIMENTS AND RESULTS

We evaluate our guardrail classifiers on both synthetic data (held-out sets and additional LLM generations) and external benchmarks focusing on other misuse categories. We also provide a deeper explanation of why our approach outperforms baseline methods, discuss metrics, calibration, and threshold selection, and analyze the trade-offs for real-world deployment.

## 4.1 DATASETS

**Synthetic Dataset.**  We use GPT 4o (2024-08-06) and carefully designed prompts to generate more than 2M (system prompt, user prompt) pairs with balanced on/off-topic labels. Examples include system prompts for specialized QA (e.g., "You are a healthcare policy Q&A bot"), summarization tasks, or other domain-specific instructions. The user prompts range from relevant domain questions to completely unrelated topics like "Write me a Python program" or attempts at discussing personal hobbies. We also inject edge cases: extremely short prompts, multi-paragraph prompts, and random multi-lingual queries. Our experiments primarily focus on a subset (roughly 17k examples) for training and validation, leaving a hold-out set for evaluation.

**External Datasets.**  We further evaluate generalization to misuse categories like jailbreaking or harmful requests by pairing these external prompts with a random system prompt from our synthetic set. Concretely, we assess:

- **JailbreakBench** (Chao et al., 2024) – A benchmark containing both benign user prompts and adversarial "jailbreak" attempts.

- **HarmBench** (Mazeika et al., 2024), **TrustLLM** (et al., 2024c), and a **Localized Harmful** dataset (Foo & Khoo, 2025) – Collections of prompts aimed at eliciting harmful or disallowed content (e.g., hate speech, extremely sensitive topics).

Although these sets target different misuse categories, from the standpoint of a specialized system prompt with a narrow domain, they are effectively "off-topic" or disallowed.

## 4.2 BASELINES

We compare with:

1. **Cosine Similarity (Embeddings).** We embed system and user prompts via bge-large-en-v1.5 (Xiao et al., 2024), compute cosine similarity, and set a heuristic threshold.

2. **K-Nearest Neighbors (6-shot).** We store embeddings for a small set of 3 on-topic and 3 off-topic examples, then classify new prompts based on nearest neighbor similarity.

3. **Pre-trained Cross-Encoder (no fine-tuning).** We use stsb-roberta-base to obtain a semantic similarity score and threshold it.

4. **Pre-trained ColBERT.** ColBERT v2 (Santhanam et al., 2022) for text relevance, thresholded for on/off-topic.

5. **Prompt Engineering Only.** We rely on instructing the LLM directly to refuse off-topic questions via system or developer instructions, with no external classifier.

6. **LLM Zero-Shot Classification.** We query a smaller LLM to label the user prompt as off-topic or on-topic, relying purely on zero-shot reasoning.

## 4.3 METRICS AND THEIR RELEVANCE

We use the following metrics:

- **ROC-AUC (Receiver Operating Characteristic Area Under Curve).** Reflects how well the model separates off-topic from on-topic examples across all possible thresholds.

- **Precision.** Fraction of predicted off-topic prompts that truly are off-topic, important for minimizing false positives (over-refusals).

$$\text{Precision} = \frac{\text{True Positives}}{\text{True Positives} + \text{False Positives}}$$

- **Recall (Sensitivity).** Fraction of true off-topic examples that are caught by the model. Important to ensure we do not allow many off-topic prompts through.

$$\text{Recall} = \frac{\text{True Positives}}{\text{True Positives} + \text{False Negatives}}$$

- **F1 Score.** The harmonic mean of precision and recall, balancing false positives and false negatives:

$$\text{F1} = 2 \times \frac{\text{Precision} \times \text{Recall}}{\text{Precision} + \text{Recall}}$$

- **Calibration.** We also measure how well predicted probabilities match empirical off-topic frequencies, important for threshold-based deployment.

These metrics are well-suited for a safety-critical classification task where both under- and over-blocking can undermine user trust and compliance.

## 4.4 PERFORMANCE ANALYSIS AND REASONS FOR IMPROVEMENT

**Results on Synthetic Data.** Table 1 shows results on a held-out set (17k examples) from our GPT 4o synthetic dataset. Both fine-tuned models outperform baselines in terms of F1 and ROC-AUC. Zero-shot LLM classification also performs well but is prone to more false positives, creating potential user frustration or system friction. We also evaluate our models on other held-out sets generated by different LLMs, including Gemini Pro 1.5, Claude 3.5 Sonnet, and Llama 3.1 405B, and observe consistent performance (see Annex 7).

**Why the Improvement?** Our fine-tuned models benefit from:

1. **Task-Specific Synthetic Data:** The classifier sees many diverse on/off-topic pairs, learning domain-invariant signals of relevance (e.g., semantic overlap, stylistic cues).

Table 1: Performance on Synthetic Data (N=17,201). We report ROC-AUC, F1, Precision, and Recall.

| Approach | Model | ROC-AUC | F1 | Precision | Recall |
|---|---|---|---|---|---|
| **Fine-tuned cross-encoder** | stsb-roberta-base | **0.99** | **0.99** | **0.99** | **0.99** |
| **Fine-tuned bi-encoder** | jina-embeddings-v2-small-en | 0.99 | 0.97 | 0.99 | 0.95 |
| Cosine similarity | bge-large-en-v1.5 | 0.89 | 0.59 | 0.97 | 0.42 |
| KNN (6-shot) | bge-large-en-v1.5 | 0.90 | 0.75 | 0.94 | 0.63 |
| Pre-trained cross-encoder | stsb-roberta-base | 0.73 | 0.68 | 0.53 | 0.93 |
| Pre-trained ColBERT | ColBERT v2 | 0.78 | 0.72 | 0.72 | 0.73 |
| Prompt engineering | GPT 4o (2024-08-06) | - | 0.95 | 0.94 | 0.97 |
| Prompt engineering | GPT 4o Mini (2024-07-18) | - | 0.91 | 0.85 | 0.91 |
| Zero-shot classification | GPT 4o Mini (2024-07-18) | 0.99 | 0.97 | 0.95 | 0.99 |

2. **Focused Architecture:** Fine-tuning cross-encoders or bi-encoders captures deeper relationships between $S$ and $U$.

3. **Balanced and Large-Scale Synthetic Set:** Synthetic generation allows covering extreme and edge cases (very short prompts, multi-lingual, adversarial style) more comprehensively than a small human-curated set could.

**Calibration.** In safety applications, well-calibrated probabilities are vital for setting thresholds. Figure 4 shows a reliability diagram for the fine-tuned cross-encoder on the synthetic hold-out set. The probabilities align well with observed frequencies, especially in high-confidence regions.

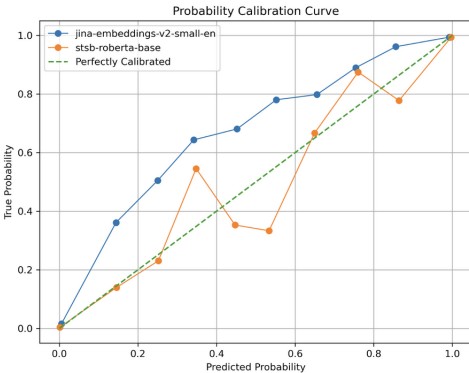

Figure 4: Calibration Plot on the Synthetic Hold-Out Set. A near-diagonal plot indicates good probability calibration.

### 4.5 GENERALIZATION TO JAILBREAK AND HARMFUL PROMPTS

To investigate broader misuse detection, we pair prompts from external datasets with random specialized system prompts. Table 2 shows results on JailbreakBench, which has both benign and adversarial user prompts. Our fine-tuned bi-encoder and cross-encoder detect a high fraction of jailbreak attempts (off-topic for a specialized system prompt), achieving better recall than naive baselines.

Table 2: Binary Classification on JailbreakBench. We list ROC-AUC, F1, Precision, and Recall.

| Approach | Model | ROC-AUC | F1 | Precision | Recall |
|---|---|---|---|---|---|
| Fine-tuned cross-encoder | stsb-roberta-base | 0.80 | 0.72 | 0.76 | 0.68 |
| Fine-tuned bi-encoder | jina-embeddings-v2-small-en | **0.92** | **0.83** | **0.84** | **0.82** |

For HarmBench, TrustLLM, and a localized dataset of harmful prompts (Foo & Khoo, 2025), Table 3 reports recall, as these datasets focus mainly on harmful content (positive class). Our mod-

els exhibit strong recall, effectively blocking harmful requests. Notably, the fine-tuned bi-encoder achieves particularly high recall on HarmBench and TrustLLM.

Table 3: Recall on HarmBench, TrustLLM, and a Localized Harmful Dataset. (These sets mostly contain only harmful prompts, so we focus on whether the guardrail detects them as off-topic.)

| Dataset | Approach | Model | Recall |
|---|---|---|---|
| HarmBench | Fine-tuned cross-encoder | stsb-roberta-base | 0.83 |
| | Fine-tuned bi-encoder | jina-embeddings-v2-small-en | **0.99** |
| TrustLLM | Fine-tuned cross-encoder | stsb-roberta-base | 0.78 |
| | Fine-tuned bi-encoder | jina-embeddings-v2-small-en | **0.97** |
| Localized harmful | Fine-tuned cross-encoder | stsb-roberta-base | 0.74 |
| | Fine-tuned bi-encoder | jina-embeddings-v2-small-en | **0.86** |

### 4.6 INFERENCE SPEED AND FEASIBILITY

Table 4 reports throughput on an NVIDIA Tesla T4. Both the bi-encoder and cross-encoder can process thousands of prompt pairs per minute, sufficient for many real-time applications.

Table 4: Inference Speed (Prompt Pairs per Minute) on an NVIDIA Tesla T4 GPU.

| Approach | Model | Pairs/min | Latency (s/pair) |
|---|---|---|---|
| Fine-tuned bi-encoder | jina-embeddings-v2-small-en | 2216 | 0.027 |
| Fine-tuned cross-encoder | stsb-roberta-base | 1919 | 0.031 |

## 5 DISCUSSION

Our experiments demonstrate that synthetic data generation, combined with fine-tuned embedding or cross-encoder classifiers, is highly effective for detecting off-topic prompts. We now discuss key limitations, possible solutions, deployment considerations, and the method's broader applicability.

### 5.1 LIMITATIONS AND FUTURE WORK

**Synthetic Data Bias.** LLMs may introduce distributional biases or style artifacts when generating synthetic data. Although we mitigate this via temperature tuning and random seed words, real-world usage might differ significantly. Future work includes incorporating active learning once real user data arrives, progressively retraining or updating the classifier.

**Breadth of System Prompts.** If the system prompt is extremely open-ended (e.g., "Chat about anything"), the notion of off-topic becomes less meaningful. Our approach is best suited to LLM applications with well-defined tasks or domains. Handling highly unstructured or unlimited tasks is an open research challenge.

**Language and Cultural Contexts.** Our experiments primarily focus on English. Different languages or cultural nuances may require specialized data generation or adaptation. Future work can extend to multilingual settings or incorporate domain-specific or cultural context more carefully.

**Towards More Complex Real-World Scenarios.** Though we demonstrated generalization to jailbreak and harmful prompts, additional complexities arise in real deployments, such as multi-turn dialogues, code generation, or multimodal inputs. We plan to extend our approach to multi-turn conversation guardrails, ensuring that off-topic detection remains robust across longer context windows.

## 6 DEPLOYMENT CONSIDERATIONS

This guardrail development methodology has been used within Government Technology Agency over the past year to build an internal suite of guardrails for various LLM applications. Specifi-

cally for off-topic guardrails, it has been deployed internally since September 2024. Moreover, the methodology has also been utilized to detect system prompt leakage or other categories of undesired model outputs. Our key considerations have been:

**Actionability and Threshold Tuning.** For real deployments, system owners can choose a threshold $t \in [0, 1]$ on the off-topic probability. A low threshold catches more off-topic prompts but may over-refuse. A high threshold reduces false positives but risks letting off-topic prompts slip by. In practice, organizations often perform pilot tests or active learning with real user data to fine-tune $t$. The well-calibrated scores from our model make such tuning more reliable. From our internal user studies, typical threshold values range between $0.4$ and $0.6$, balancing user satisfaction and compliance.

**Integration with Alignment.** Guardrails complement alignment by providing an external filter that is easier to update. If a specific off-topic scenario emerges post-deployment (e.g., new domain requests), the guardrail classifier can be rapidly retrained or updated with minimal changes to the underlying LLM.

**Model Choice.** For longer system prompts, the bi-encoder can handle more tokens with slightly lower compute cost, whereas the cross-encoder typically offers higher accuracy for shorter text. A hybrid approach is also possible: run a faster embedding-based approach, then a cross-encoder re-check if uncertain.

**Active Learning Pipeline.** Once the system is live, real user queries can be sampled and quickly labeled. Incremental retraining with this real data helps address distribution shifts and correct synthetic data biases.

## 7 CONCLUSION

We presented a flexible, data-free methodology for developing guardrails for LLMs, applied specifically to off-topic prompt detection. By systematically defining the problem space and leveraging LLMs to generate large, diverse synthetic datasets, we trained high-performance classifiers that both reduce false positives and generalize to other misuse categories (e.g., jailbreak or harmful prompts). Our open-source release of the synthetic dataset and trained guardrail models aims to accelerate broader research and practical deployments in LLM safety.

Nevertheless, important limitations remain. Synthetic data may not fully represent real-world behaviors, especially in multilingual or high-context scenarios, and extremely open-ended system prompts pose conceptual challenges to off-topic detection. As future work, we intend to explore deeper domain adaptation, multi-turn dialogues, and extended language coverage. We hope that our approach and open-source contributions will foster more robust and adaptive guardrail solutions, ensuring safer and more trustworthy LLM deployments across diverse applications.

All authors contributed to conceptualizing the project and designing the methodology. Gabriel and Shing Yee led the implementation and experiments, and Gabriel wrote the manuscript, with input and revisions from all co-authors.

We thank our colleagues at Government Technology Agency, especially at the AI Practice and Responsible AI team, for feedback during the early stages of this work. We are also grateful to the open-source community for providing pre-trained models and helpful software libraries.

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

## A ANNEX: ADDITIONAL SYNTHETIC DATA EVALUATION

Table 5: Performance on Synthetic Data Generated by Gemini Pro 1.5 and Claude 3.5 Sonnet (N=326).

| Approach | Model | ROC-AUC | F1 | Precision | Recall |
|---|---|---|---|---|---|
| Fine-tuned cross-encoder | stsb-roberta-base | 0.99 | 0.99 | 0.99 | 0.99 |
| Fine-tuned bi-encoder | jina-embeddings-v2-small-en | 0.99 | 0.99 | 0.99 | 0.99 |

Table 6: Performance on Synthetic Data Generated by Llama 3.1 405B (N=29,635).

| Approach | Model | ROC-AUC | F1 | Precision | Recall |
|---|---|---|---|---|---|
| Fine-tuned cross-encoder | stsb-roberta-base | 0.99 | 0.96 | 0.97 | 0.94 |
| Fine-tuned bi-encoder | jina-embeddings-v2-small-en | 0.99 | 0.99 | 0.97 | 0.90 |

