# OpenReview forum: "A Flexible Large Language Models Guardrail Development Methodology Applied to Off-Topic Prompt Detection"
_ICLR.cc/2025/Workshop/BuildingTrust — BuildingTrust_

### Official Review · Reviewer_K8o5 · 2025-02-16
**Review for "A Flexible Large Language Models Guardrail Development Methodology Applied to Off-Topic Prompt Detection"**

**Rating:** 7
**Confidence:** 4

**Review:**

# Summary
The paper presents a practical methodology for developing LLM guardrails focused on detecting off-topic user prompts. The approach leverages synthetic data generation via LLMs to train off-topic classifiers without requiring real-world data. The authors compare bi-encoder and cross-encoder models, demonstrating high performance on synthetic benchmarks and generalization to jailbreak and harmful prompt detection. The paper also releases synthetic datasets and trained models to support the broader research community.

## Strengths
- The paper addresses a relevant safety problem in LLM deployment—off-topic misuse—that is often underexplored compared to harmful content.
- The data-free guardrail development methodology is a pragmatic approach to overcome pre-deployment data scarcity.
- Strong empirical comparisons with baseline classifiers and generalization assessments on jailbreak and harmful prompt benchmarks.
- The dataset and models contribute to the development of LLM safety solutions in real-world settings.

## Weaknesses
- Results are primarily based on synthetic data, with minimal real user data evaluation. The distribution gap between synthetic and real-world usage remains a concern.
- Off-topic detection closely resembles domain relevance classification, reducing theoretical novelty.
- Insufficient analysis of edge cases and borderline prompts in deployment contexts.

---

### Official Review · Reviewer_PTXv · 2025-02-20
**Good paper**

**Rating:** 8
**Confidence:** 4

**Review:**

Pro:

Novel Methodology
The proposed approach eliminates the need for real-world data in the pre-deployment phase by using synthetic data generation, which is both innovative and practical.

Comprehensive Framework
The paper clearly outlines a three-step guardrail development framework: qualitative problem analysis, synthetic data generation via LLM prompting, and model training.
This systematic approach ensures that edge cases, diversity in prompt styles, and domain-specific challenges are well addressed.


Generalization and Deployment Considerations
The method not only improves off-topic detection but also generalizes to other misuse cases (e.g., jailbreak and harmful prompts).
Detailed discussion on deployment issues—such as threshold tuning, integration with alignment methods, model choice, and active learning pipelines—shows strong practical relevance. (But please notice that the jailbreak or harmful request are easily rejected by the aligned model, maybe it can be better to test them in uncensored model)

Could improve:

Reliance on Synthetic Data
While synthetic data generation is a powerful idea, the paper acknowledges that such data may not fully capture the nuances of real-world user behavior, especially in multilingual or high-context scenarios.
Future work could explore integrating real user data (through active learning) to further refine and validate the guardrail classifiers.

---

### Official Review · Reviewer_XGDX · 2025-02-28
**In this work authors work on off-topic detection by showing a methodology of generating data and later using it to create off-topic classifier. This method can be used with chat applications that have a specific topic designed for them allowing for light-weight detector of off-topic queries.**

**Rating:** 6
**Confidence:** 4

**Review:**

This work is well written, and the idea of off-topic detection seems like a good research direction for LLMs developed with specific topics in mind.

Pros:
* The authors show how one could use their method in an end-to-end manner, starting from the creation of the data and obtaining an off-topic classifier.
* Those classifiers are lightweight, making them useful in real-time applications.
Cons:
* We can observe a large drop in performance when this method is applied to other datasets, potentially highlighting its problem with generalization and possible overfitting.
* More evaluations on data that don't come from the same distribution from which training data was generated would better show how well this method would generalize. The authors could have tried generating an eval set with another LLM.
* I would like the authors to provide hyperparameters used for training.
* L259 missing citation?
* L314 missing reference to Appendix

---

### Decision · Program_Chairs · 2025-03-04

**Decision:**

Accept

**Comment:**

The paper's reliance on synthetic data raises concerns about its ability to generalize to real-world user behavior, particularly in multilingual or high-context scenarios. Additionally, the method exhibits significant performance drops when applied to different datasets, suggesting potential overfitting and a lack of robustness, which could have been better evaluated using data from another LLM or real user interactions.